# A figure of merit for efficiency roll-off in TADF-based organic LEDs

S. Diesing[1,2,3], L. Zhang[1,2,3], E. Zysman-Colman[2✉] & I. D. W. Samuel[1,3✉]

Organic light-emitting diodes (OLEDs) are a revolutionary light-emitting display technology that has been successfully commercialized in mobile phones and televisions[1,2]. The injected charges form both singlet and triplet excitons, and for high efficiency it is important to enable triplets as well as singlets to emit light. At present, materials that harvest triplets by thermally activated delayed fluorescence (TADF) are a very active field of research as an alternative to phosphorescent emitters that usually use heavy metal atoms[3,4]. Although excellent progress has been made, in most TADF OLEDs there is a severe decrease of efficiency as the drive current is increased, known as efficiency roll-off. So far, much of the literature suggests that efficiency roll-off should be reduced by minimizing the energy difference between singlet and triplet excited states ($\Delta E_{ST}$) to maximize the rate of conversion of triplets to singlets by means of reverse intersystem crossing ($k_{RISC}$)[5–20]. We analyse the efficiency roll-off in a wide range of TADF OLEDs and find that neither of these parameters fully accounts for the reported efficiency roll-off. By considering the dynamic equilibrium between singlets and triplets in TADF materials, we propose a figure of merit for materials design to reduce efficiency roll-off and discuss its correlation with reported data of TADF OLEDs. Our new figure of merit will guide the design and development of TADF materials that can reduce efficiency roll-off. It will help improve the efficiency of TADF OLEDs at realistic display operating conditions and expand the use of TADF materials to applications that require high brightness, such as lighting, augmented reality and lasing.

Organic light-emitting diodes (OLEDs) are now widely used in displays and are being developed for applications in lighting, sensing and communications[1,2]. They consist of layers of charge transporting and light-emitting organic semiconductors in between two electrodes, at least one of which is transparent. When the injected charges recombine, they form both singlet and triplet excitons. Spin statistics suggest three triplets form for each singlet, a ratio that has been verified for evaporated OLEDs using low molecular weight emitters[21]. In OLEDs using fluorescent materials, only the singlets emit light. Phosphorescent OLED materials were therefore developed to obtain light emission from the triplets as well[22]. These work very well for red and green emission, but there is not yet a blue phosphorescent emitter meeting all commercial requirements[23]. Consequently, there is currently great interest in thermally activated delayed fluorescence (TADF) as an alternative approach to obtaining light from triplets[3,4]. Following the pioneering work of Adachi and coworkers in 2011, there have been more than 4,000 papers with the keyword thermally activated delayed fluorescence[24,25] (based on results from 16 February 2024 that mention thermally activated delayed fluorescence or TADF since 2011).

A problem in both organic and inorganic LEDs is that as the drive current is increased for more light output, the efficiency decreases[26]. This is known as efficiency roll-off and is illustrated in Fig. 1a, which shows the efficiency as a function of current density for prototypical examples of fluorescent, phosphorescent and TADF OLEDs[3,27,28].

Figure 1a shows that the phosphorescent and TADF OLEDs have more than four times the efficiency of the fluorescent OLEDs, but that their efficiency decreases and particularly severely for the TADF OLEDs as the current density is increased. To compare the behaviour of a wide range of OLEDs of each type, we define $J_{90}$ as the current density at which the external quantum efficiency (EQE) falls to 90% of its peak value, as illustrated in Fig. 1b.

We have extracted $J_{90}$ from published data on a wide range of OLEDs together with their EQE at a practical luminance of 1,000 cd m$^{-2}$. These are plotted for each class of OLEDs and each colour in Fig. 1c. The ideal behaviour would be high $J_{90}$ (for low efficiency roll-off) and high EQE: that is, the top right quadrant of the graph. Most fluorescent OLEDs fall in the green rectangle (A), which is a region of high $J_{90}$ but low EQE. Most phosphorescent OLEDs (and a few others) fall in the blue rectangle (B). The upper half of this rectangle represents OLEDs with high efficiency and fairly high $J_{90}$. TADF OLEDs fall mainly in region C. Notably, there is a much wider spread of both EQE and $J_{90}$ than for the other classes of OLEDs, possibly because TADF OLEDs is a much younger field. The upper right part of region C shows that there are some reports of TADF OLEDs with high EQE and moderately high $J_{90}$, although lower than for good phosphorescent devices. However, region C also extends to extremely low values of $J_{90}$ that is, there are many TADF devices experiencing significant efficiency roll-off at current densities below 0.1 mA cm$^{-2}$. Even for a green device, this would correspond

[1]Organic Semiconductor Centre, SUPA, School of Physics and Astronomy, University of St Andrews, St Andrews, UK. [2]Organic Semiconductor Centre, EaStCHEM, School of Chemistry, St Andrews, UK. [3]These authors contributed equally: S. Diesing, L. Zhang, I. D. W. Samuel. ✉e-mail: eli.zysman-colman@st-andrews.ac.uk; idws@st-andrews.ac.uk

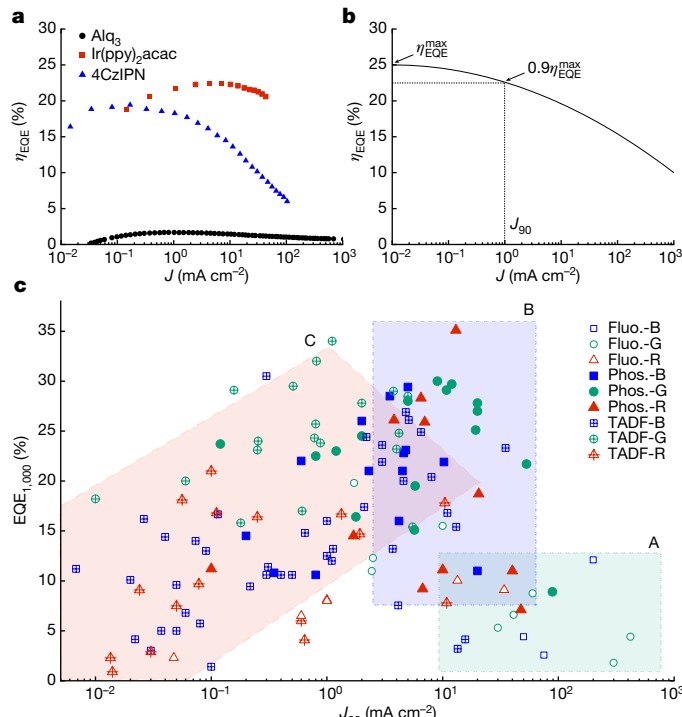

**Fig. 1 | Examples of efficiency roll-off. a,** Efficiency roll-off of prototypical fluorescent ($Alq_3$), phosphorescent ($Ir(ppy)_2acac$) and TADF (4CzIPN) OLEDs[3,27,28]. **b,** Schematic graph showing definition of $J_{90}$. **c,** Graph of the relation between $J_{90}$ and $EQE_{1,000}$ for TADF[5,38,39,48–113], fluorescent (Fluo.)[28,114–129] and phosphorescent (Phos.)[86–88,128–155] devices emitting in the red (R), green (G) and blue (B) regions of the spectrum (for the references, see Methods).

to a brightness of at most 100 cd $m^{-2}$, whereas typical displays run at 400 cd $m^{-2}$ and their individual pixels often run at much higher brightness to achieve an average 400 cd $m^{-2}$ on the display[29]. Hence, many reported TADF OLEDs have severe efficiency roll-off and even the best have significant efficiency roll-off ($J_{90}$ of a few mA $cm^{-2}$).

## Efficiency roll-off in TADF devices

This brings us to the central question of this analysis, which is, what can be done in terms of emitter design to reduce efficiency roll-off (that is, increase $J_{90}$)? In other words, which photophysical processes need to be tuned by molecular design to minimize inherent limitations of the emitter that contribute to efficiency roll-off? Efficiency roll-off arises both from the emitter design and the device design, but for an optimized device design (for example, balanced charge carriers and wide recombination zone) it will ultimately be limited by the properties of the emitter.

To identify the crucial parameters for emitter design, we first need to consider what causes efficiency roll-off. Studies in phosphorescent OLEDs have shown that triplet–triplet annihilation (TTA) and triplet–polaron annihilation (TPA) are the main loss mechanisms as the current density is increased[29–31]. A similar understanding is developing in TADF OLEDs in which TTA, TPA and singlet–triplet annihilation (STA) may all contribute[32–34]. These are all bimolecular processes and thus much more severe at higher excitation densities. Furthermore, as all these processes involve triplets, they can be mitigated by reducing the triplet lifetime and hence reducing the triplet population. This has been achieved successfully in phosphorescent OLEDs by engineering the light-emitting material (for example, by using an iridium complex) to show a large radiative rate constant from the triplet state and thus achieving a relatively short triplet lifetime of around 1 µs.

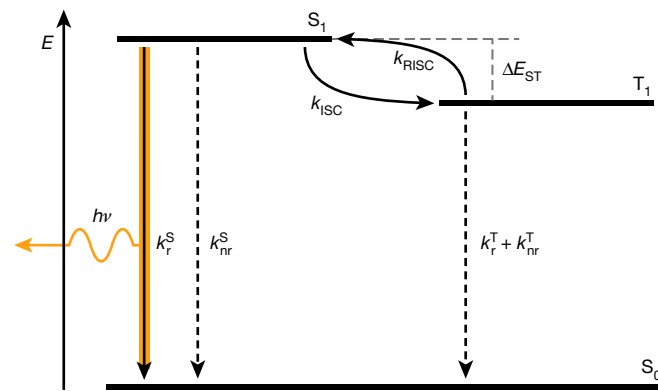

**Fig. 2 | Simplified Jablonski diagram of a TADF emitter.** The excited singlet and triplet states $S_1$ and $T_1$, respectively, are shown in equilibrium due to the occurrence of both intersystem crossing ($k_{ISC}$) and reverse intersystem crossing ($k_{RISC}$) enabled by the small energy gap ($\Delta E_{ST}$) between $S_1$ and $T_1$. A TADF OLED emits light by radiative decay from $S_1(k_r^S)$, whereas non-radiative decay from both $S_1(k_{nr}^S)$ and $T_1(k_{nr}^T)$, as well as the negligible radiative decay from $T_1(k_r^T)$ are other deactivation pathways of excited species.

For comparison, delayed fluorescence lifetimes in organic TADF materials range from 1 µs to beyond 500 µs. We briefly note that as well as reducing efficiency, bimolecular processes involving triplets are also a main mechanism of device degradation, providing a further reason to reduce the triplet population in operating devices[35].

Hence to reduce efficiency roll-off, we need to reduce triplet lifetime or, more precisely, the triplet population during device operation. This, however, is not as simply achieved as in the case of phosphorescence. The key photophysical processes in a TADF emitter are shown in Fig. 2. Singlets are converted to triplets via intersystem crossing (ISC) with rate constant $k_{ISC}$, and triplets to singlets via reverse intersystem crossing (RISC) at rate constant $k_{RISC}$. There is potentially radiative and non-radiative decay of both triplets and singlets, although in a good TADF material $k_r^S$ will be much larger than any of $k_{nr}^S$, $k_r^T$ and $k_{nr}^T$ (refs. 36,37). The main approach advocated in the literature for reducing efficiency roll-off is to increase $k_{RISC}$, commonly by reducing the energy difference between singlet and triplet excited states ($\Delta E_{ST}$) through molecular design by reducing the exchange integral between the highest occupied and lowest unoccupied molecular orbitals. In addition, there have been other attempts to increase $k_{RISC}$, for example by the use of heavy atoms, to increase spin–orbit coupling (SOC)[5,38]. The emphasis on $k_{RISC}$ is so strong that since 2016 there have been 16 publications in the *Nature* family alone exploring $k_{RISC}$ (refs. 5–20). However, the expected improvement in $J_{90}$ has not always materialized.

To understand how $J_{90}$ depends on $k_{RISC}$, we have plotted the graph shown in Fig. 3. There is some correlation (Spearman correlation $\rho = 0.638$) in so far as there is a tendency towards higher $J_{90}$ for higher $k_{RISC}$, but there is an enormous spread of the data (considering this is a log–log plot). For example, the blue dashed rectangle shows that $J_{90}$ of roughly 2 mA $cm^{-2}$ can be achieved with $k_{RISC}$ from 2 to $20 \times 10^5$ $s^{-1}$. The insufficiency of $k_{RISC}$ as a guide for molecular design is vividly demonstrated by the red dashed rectangle that shows $J_{90}$ for molecules designed with a high $k_{RISC}$ of $8–15 \times 10^5$ $s^{-1}$. The values of $J_{90}$ range from 0.03 to 40 mA $cm^{-2}$, that is, by more than three orders of magnitude, showing $k_{RISC}$ alone is inadequate as a predictor of efficiency roll-off.

## Derivation of FOM

To develop guidelines for TADF materials design to reduce efficiency roll-off in OLEDs, we should first look more closely at Fig. 2 and the mechanism of TADF.

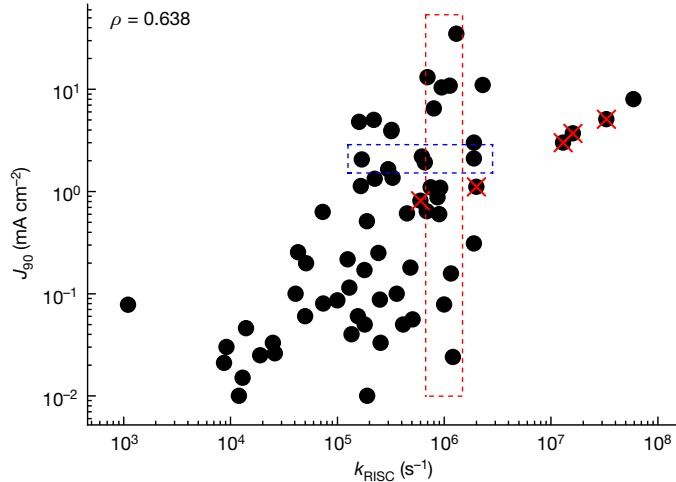

**Fig. 3 | Data analysis.** $J_{90}$ of the reported TADF OLEDs with respect to $k_{RISC}$ (Spearman correlation $\rho = 0.638$). Red crosses present TADF molecules containing heavy atoms that benefit from enhanced SOC to increase $k_{RISC}$. Data inside the dashed boxes are for comparison.

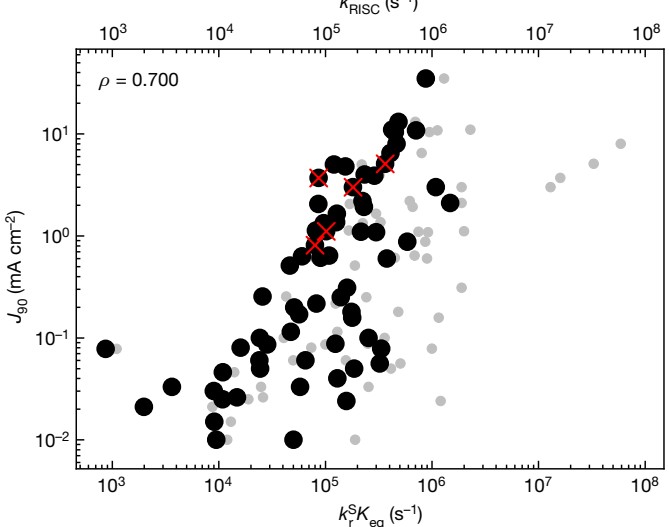

**Fig. 4 | FOM.** Correlations between $J_{90}$ and $k_r^S K_{eq}$ with a Spearman correlation coefficient of $\rho = 0.700$. Red crosses identify TADF molecules containing heavy atoms that enhance SOC, which leads to an increase in $k_{RISC}$. In grey circles, the correlation of $J_{90}$ and $k_{RISC}$ from Fig. 3 is shown for comparison.

The physics of TADF is often studied using transient photoluminescence (PL) measurements in which the emitter is excited using a laser pulse. Excitons are generated in the excited singlet state ($S_1$), and the decay of the excited state is slowed by cycling to and from the triplet state ($T_1$) by ISC and RISC, respectively. In an OLED, charge injection leads to a buildup of both $S_1$ (25%) and $T_1$ (75%) excitons as well as polarons. There is a dynamic equilibrium between $S_1$ and $T_1$ facilitated by the ISC/RISC cycling. To ascertain on which side the dynamic equilibrium lies, an equilibrium constant $K_{eq}$ is defined as

$$K_{eq} = \frac{[S_1]}{[T_1]}. \qquad (1)$$

For a three-level TADF OLED under low constant current electrical excitation, the equilibrium constant is given as follows (see Methods for the derivation)

$$K_{eq} = \frac{3k_{RISC} + k^T}{3k^S + k_{ISC}}, \qquad (2)$$

where $k^S$ is the sum of the rate constants for ISC ($k_{ISC}$), radiative ($k_r^S$) and non-radiative ($k_{nr}^S$) decay from $S_1$, and $k^T$ is the sum of the rate constants for RISC ($k_{RISC}$), radiative ($k_r^T$) and non-radiative ($k_{nr}^T$) decay from $T_1$. As explained earlier, to minimize the EQE roll-off a low $T_1$ population is necessary to suppress TTA and to a lesser extent STA and TPA. For an OLED operated at high brightness this translates to the requirement of maximizing the $S_1$ population relative to the $T_1$ population, which can be achieved by maximizing $K_{eq}$. Furthermore, according to Le Chatelier's principle, an equilibrium can be moved to a desired product by removing the product from the equilibrium. Here, the radiative decay of $S_1$ excitons is the desired product. Therefore, to minimize the fraction of triplet excitons in the steady-state OLED emitters should be developed (or selected) to maximize the product of radiative rate constant and equilibrium constant. In a good OLED, nearly all electrically excited excitons decay radiatively, that is $k_{nr}^S = k_{nr}^T = 0$. Thus, for a TADF emitter with photoluminescence quantum yield near unity and no phosphorescence contribution ($k_r^T = 0$), which is reasonable for good organic emitters, a figure of merit (FOM) for efficiency roll-off can be formulated as

$$k_r^S K_{eq} = \frac{4k_r^S k_{RISC}}{3k_r^S + 4k_{ISC}}. \qquad (3)$$

Figure 4 shows $J_{90}$ plotted as a function of this FOM. There is a stronger correlation with $J_{90}$ ($\rho = 0.700$) than $k_{RISC}$ with $J_{90}$ ($\rho = 0.638$). Higher $k_r^S K_{eq}$ leads to higher $J_{90}$. Accordingly, maximizing $k_r^S K_{eq}$ and thus minimizing the $T_1$ population under electrical excitation is a better strategy for improving efficiency roll-off than considering $k_{RISC}$ alone. Figure 4 compares efficiency roll-off as a function of our FOM (black circles) with efficiency roll-off as a function of $k_{RISC}$ (small grey circles). The FOM has a narrower spread of values as would be expected for the improved correlation.

It is interesting to apply this FOM to recent attempts to increase $k_{RISC}$ by incorporating heavy atoms into the molecule to increase SOC[5,38]. These studies are shown by red crosses in Figs. 3 and 4. This strategy is broadly successful at leading to fast $k_{RISC}$ but does not necessarily lead to the highest $J_{90}$ as $k_{ISC}$ also increases, or $k_r^S$ decreases. This interplay between these parameters is captured by the FOM as can be seen from the red crosses in Fig. 4 being in the same region as other materials. At the same time, incorporation of these larger atoms that result in weaker bonds also leads to faster non-radiative pathways and potentially poor device stability. Here, we can see that $k_{RISC}$ and the proposed FOM give distinct assessments of the heavy atom approach, and that the latter is a better predictor of $J_{90}$ for future molecular design. Another possible guide for design is (as for phosphorescent devices) short delayed fluorescence lifetime ($\tau_{DF}$)[39]. The correlation of $J_{90}$ with $\tau_{DF}$ is shown in Extended Data Fig. 1a. There is a good correlation ($\rho = -0.685$), although still some scatter. Actually, $\tau_{DF}$ has a much stronger correlation with the proposed FOM ($\rho = -0.801$) than with $k_{RISC}$ ($\rho = -0.709$). In other words, the proposed FOM not only predicts the efficiency roll-off, but also clarifies the key physical processes that need to be optimized to achieve low efficiency roll-off. Although measuring $\tau_{DF}$ would be an effective way of screening materials for low potential efficiency roll-off after they have been synthesized, our FOM gives more insight into how to design a material for low efficiency roll-off by showing the exact combination of rate constants that should be optimized.

For many TADF materials $k_{ISC}$ is substantially faster than $k_r^S$, in which case the FOM can be simplified to

$$k_r^S K_{eq} \approx \frac{k_r^S k_{RISC}}{k_{ISC}} \text{ for } k_r^T = k_{nr}^T = k_{nr}^S = 0 \text{ and } k_{ISC} \gg k_r^S \qquad (4)$$

This simplified FOM highlights the competition between $k_{ISC}$, $k_{RISC}$ and $k_r^S$ very clearly. It is equivalent to $k_r^S K_{eq}$ in the regime where $k_r^S$ is smaller than $k_{ISC}$ (Extended Data Fig. 2).

## Other factors affecting efficiency roll-off

Although there is a good correlation between $J_{90}$ and the proposed FOM, there is a significant spread of data points in Fig. 4. This can be understood to arise because efficiency roll-off involves a combination of the intrinsic properties of the emitting molecule with the extrinsic properties of the device. An analogous situation exists when using photoluminescence quantum yield as a predictor of device efficiency: whether a material realizes its full potential also depends on the device. Similarly, our FOM describes the best that could be achieved with a particular light-emitting material in a device limited by the triplet population. In real devices, many factors, especially imperfect charge balance, could lead to worse performance than this ideal case, and hence can explain the spread of the data in Fig. 4. In addition, at low current density some devices show efficiency increasing with current density, as can be seen for the devices in Fig. 1a. As $J_{90}$ is taken as a reduction from peak efficiency, this will lead to higher values of $J_{90}$ than in devices with peak efficiency at very low current. There is another example of this effect in Extended Data Fig. 3 that compares two 2CzPN devices[3,40]. It should also be noted that practice for determining rate constants varies[37,41], which could also contribute to the spread.

Another important factor that could contribute to the spread of data is that the effect of a given triplet population depends on the material. In particular, reported TTA rate constants $\gamma_{TT}$ are widely spread over eight orders of magnitude ($10^{-18}$–$10^{-10}$ cm$^3$ s$^{-1}$)[32,33,42,43]. So, increasing the FOM will reduce triplet population, and is beneficial (increases $J_{90}$) but the improvement arising from the reduced triplet population depends on the value of $\gamma_{TT}$. Similar considerations apply to STA, in which again there is a range of $\gamma_{ST}$, and the relative importance of STA and TTA depends on the relative values of $\gamma_{TT}$ and $\gamma_{ST}$. As these rate constants are not yet widely measured, we have not at this stage attempted to incorporate them into a FOM. However, we show their potential effect in Extended Data Fig. 4, which shows calculations of how $J_{90}$ would depend on FOM for systems with $k_r^S$ between $10^5$ and $10^{10}$ s$^{-1}$, $k_{ISC}/k_r^S$ between $10^{-1}$ and $10^3$, and $k_r^S K_{eq}$ between $10^2$ and $10^8$ s$^{-1}$ for a range of values of $\gamma_{ST}$, $\gamma_{TT}$ and $k_r^S$. Extended Data Fig. 3a shows how for a given FOM, each order of magnitude change in $\gamma_{TT}$ leads to an order of magnitude change in $J_{90}$. Extended Data Fig. 4b shows the potential interplay between TTA and STA. The $J_{90}$ value behaves nearly linearly with $k_r^S K_{eq}$ when only TTA is considered (red dots, the slope is 2). If only STA is considered, there is still a correlation; however, at the same FOM, higher $J_{90}$ is achieved when $k_r^S$ is large. If both TTA and STA are significant, then the efficiency is limited by TTA at low FOM and by $k_r^S$ at high FOM.

We also note that the kinetics of thin films can result in a multi-exponential transient PL, which is caused by conformational disorder[44]. Such a decay can be analysed using a Laplace transformation of the three-level kinetics of each conformer[45]. Our analysis does not include conformational disorder but could be applied in a similar manner to the analysis of multi-exponential transient PL caused by conformational disorder.

## Conclusion

Our analysis has important implications for the rapidly growing field of TADF OLED development. At present, many such devices suffer such severe efficiency roll-off that they are unsuitable for practical application and, as we have shown, current emitter design focusing on maximizing $k_{RISC}$ alone is not an effective strategy. On the basis of the insight from considering the quasi-equilibrium in TADF, we instead propose that the focus of materials design and development should

shift to maximizing a FOM that combines the physical processes that determine efficiency roll-off. Target values of the FOM will depend on the requirements of particular applications, as well as device design and severity of bimolecular effects. We estimate values of FOM required for materials with chromaticity close to the BT2020 standard[46], and with Gaussian emission spectra of width 15 nm in the blue, 30 nm in the green and 45 nm in the red. We use the calculation for Extended Data Fig. 4a with $\gamma_{TT} = 10^{-13}$ cm$^3$ s$^{-1}$ and find the FOM required to achieve 90% of a peak EQE of 25% at a brightness of 1,000 cd m$^{-2}$. We find that for a deep blue emitter ($\lambda_{max} = 467$ nm, CIE 1931 colour space (0.131, 0.049)) a FOM of at least $1.5 \times 10^5$ s$^{-1}$ is required. For a green emitter ($\lambda_{max} = 529$ nm, CIE (0.169, 0.772)) a FOM of $5.1 \times 10^4$ s$^{-1}$ would be required, and for red ($\lambda_{max} = 650$ nm, CIE (0.708, 0.292)) an FOM of at least $1.3 \times 10^5$ s$^{-1}$ is needed.

In terms of material design for low efficiency roll-off, it is not necessary to maximize $k_{RISC}$, but it is very desirable to maximize $k_{RISC}$ relative to $k_{ISC}$ (without sacrificing $k_r^S$). It is also a useful strategy to seek materials with high $k_r^S$ (providing $k_{RISC}/k_{ISC}$ is not reduced), which is also the underlying physics for hyperfluorescent OLEDs[47], where the rate constant of Förster resonance energy transfer takes the place of $k_r^S$ in the FOM and lowers the triplet population on the TADF sensitizer. At the same time, there is a need to understand which process dominates the efficiency roll-off. Whereas all main annihilation processes scale with the triplet population and thus inversely with our proposed FOM, the relative importance of these processes in each OLED is not sufficiently known. Therefore, there is a need to measure both $\gamma_{ST}$ and $\gamma_{TT}$ in a wider set of devices to fully understand how the excited-state kinetics of the emitter need to be engineered to reduce efficiency roll-off.

We hope that our FOM and these insights will enable the field of TADF OLEDs to overcome the challenge of efficiency roll-off and advance more rapidly to applications in displays, lighting and beyond.

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

## Methods

### Data collection

We considered the reported efficiency roll-off behaviour of TADF OLEDs published in peer-reviewed journals between 2016 and 2022. The data of the OLED and emitter were included in the analysis if the following criteria were met:

1. The reported OLED was vacuum-processed, in a bottom-emitting device structure with a TADF emitter in a host material.
2. Photophysical characterization of the thin film used as the emission layer was reported.
3. The photoluminescence quantum of the emitter film was reported to exceed 60%.
4. The calculation of all TADF rate constants was clearly detailed.
5. Device data clearly showed $J_{90}$ data or the presented device data allowed for a reasonable estimation of $J_{90}$.

Applying these criteria led to a total of 66 devices from 46 publications being included in our analysis[5,38,39,48–84,156–160].

For comparison, Fig. 1c shows the relation between $J_{90}$ and the EQE at 1,000 cd m$^{-2}$ (EQE$_{1,000}$) for TADF OLEDs[5,38,39,48–113] with representative fluorescent[28,114–129] and phosphorescent[86–88,128–155] devices across red, green and blue colours.

### Steady-state population of the excited states

The kinetics of a TADF emitter as shown in Fig. 2 under electrical excitation can be described by the rate equations for the excited singlet state ($S_1$) and the triplet state ($T_1$) when neglecting annihilation processes as follows.

$$\frac{d}{dt}[S_1]_t = -\underbrace{(k_r^S + k_{nr}^S + k_{ISC})}_{=k^S}[S_1]_t + k_{RISC}[T_1]_t + \frac{1}{4}\gamma[n]_t^2 \tag{5}$$

$$\frac{d}{dt}[T_1]_t = -\underbrace{(k_r^T + k_{nr}^T + k_{RISC})}_{=k^T}[T_1]_t + k_{ISC}[S_1]_t + \frac{3}{4}\gamma[n]_t^2 \tag{6}$$

where $k^S$ is the sum of the rate constants for ISC ($k_{ISC}$), radiative ($k_r^S$) and non-radiative ($k_{nr}^S$) decay from $S_1$, where $k^T$ is the sum of the rate constants for RISC ($k_{RISC}$), radiative ($k_r^T$) and non-radiative ($k_{nr}^T$) decay from $T_1$, and $\gamma$ is the Langevin recombination rate. The derivative of the polaron population $[n]_t$ can be sufficiently approximated by not distinguishing between the charge of the polaron as

$$\frac{d}{dt}[n]_t = \frac{J(t)}{ed} - \gamma[n]_t^2, \tag{7}$$

where $J(t)$ is the current density at time $t$, $d$ is the thickness of the emission zone and $e$ is the elementary charge.

In normal device operation, the OLED is driven at constant current density ($J(t) = J_{const}$) so the excited-state populations reach a steady state given by equation (8).

$$[n] = \sqrt{\frac{J_{const}}{\gamma ed}}. \tag{8}$$

The steady-state population of $S_1$ and $T_1$ ($[S_1]$ and $[T_1]$, respectively) can be obtained by substituting the differential equation for $S_1$ in steady state into the differential equation of $T_1$ steady state

$$\frac{d}{dt}[S_1] = -k^S[S_1] + k_{RISC}[T_1] + \frac{1}{4}\gamma[n]^2 = 0 \tag{9}$$

$$\Rightarrow [S_1] = \frac{k_{RISC}[T_1] + \frac{1}{4}\gamma[n]^2}{k^S} \tag{10}$$

$$\frac{d}{dt}[T_1] = -k^T[T_1] + k_{ISC}[S_1] + \frac{3}{4}\gamma[n]^2 = 0 \tag{11}$$

$$\Rightarrow [T_1] = \frac{k_{ISC}\frac{k_{RISC}[T_1] + \frac{1}{4}\gamma[n]^2}{k^S} + \frac{3}{4}\gamma[n]^2}{k^T} = \frac{k_{ISC}k_{RISC}}{k^S k^T}[T_1] + \left(\frac{k_{ISC}}{k^S} + 3\right)\frac{\gamma[n]^2}{4k^T} \tag{12}$$

$$\Leftrightarrow [T_1] = \frac{\left(\frac{k_{ISC}}{k^S} + 3\right)\frac{\gamma[n]^2}{4k^T}}{1 - \frac{k_{ISC}k_{RISC}}{k^S k^T}} = \frac{k_{ISC} + 3k^S}{4(k^T k^S - k_{ISC}k_{RISC})}\gamma[n]^2$$
$$= \frac{3k_r^S + 3k_{nr}^S + 4k_{ISC}}{4k^S(k_r^T + k_{nr}^T) + 4k_{RISC}(k_r^S + k_{nr}^S)}\gamma[n]^2 \tag{13}$$

$$\Rightarrow [S_1] = \frac{1}{k^S}\left(k_{RISC}[T_1] + \frac{1}{4}\gamma[n]^2\right) = \left(k_{RISC}\frac{k_{ISC} + 3k^S}{(k^T k^S - k_{ISC}k_{RISC})} + 1\right)\frac{\gamma[n]^2}{4k^S}$$
$$= \left(k_{RISC}\frac{k_{ISC} + 3k^S}{k^S(k_r^T + k_{nr}^T) + k_{RISC}(k_r^S + k_{nr}^S)} + 1\right)\frac{\gamma[n]^2}{4k^S} \tag{14}$$

By inserting equation (8) in equations (13) and (14) the steady-state populations are given as a function of the current density as

$$[S_1] = \left(k_{RISC}\frac{3k_r^S + 3k_{nr}^S + 4k_{ISC}}{k^S(k_r^T + k_{nr}^T) + k_{RISC}(k_r^S + k_{nr}^S)} + 1\right)\frac{1}{4k^S}\frac{J_{const}}{ed} \tag{15}$$

$$[T_1] = \frac{3k_r^S + 3k_{nr}^S + 4k_{ISC}}{4k^S(k_r^T + k_{nr}^T) + 4k_{RISC}(k_r^S + k_{nr}^S)}\frac{J_{const}}{ed} \tag{16}$$

### Calculation of $J_{90}$ for OLED examples

A set of 1,287 kinetic parameters for the equations (5) and (6) was generated, using the permutation of the input variables in Extended Data Table 1, with

$$k_{RISC} = k_r^S K_{eq}\left(\frac{3}{4} + \frac{k_{ISC}}{k_r^S}\right) \tag{17}$$

and a thickness of the emission zone of $d = 10$ nm, a Langevin recombination rate[161] of $\gamma = 6.8 \times 10^{-17}\frac{m^3}{s}$ as well as all other rate constants set to 0.

For the calculation of Extended Data Fig. 3a,b, the bimolecular rate constants were set to the values shown in the figure. $J_{90}$ was obtained by minimizing equation (18) using the python package scipy[162].

$$(\bar{\eta}_{EQE} - 0.9)^2 = \left(\frac{\eta_{EQE}(J)}{\eta_{EQE}^0} - 0.9\right)^2 = \left(\frac{\eta_{IQE}(J)}{\eta_{IQE}^0} - 0.9\right)^2 \tag{18}$$

where $\eta_{IQE}(J)$ is the internal quantum efficiency (IQE) at current density $J$ considering annihilation processes and $\eta_{IQE}^0$ is the IQE without considering annihilation processes. Both are given by

$$\eta_{IQE} = (k_r^S[S_1] + k_r^T[T_1])\frac{\gamma d}{J} \tag{19}$$

For $\eta_{IQE}^0$, the singlet population $[S_1]$ and triplet population $[T_1]$ were obtained from equations (15) and (16), respectively.

For $\eta_{IQE}(J)$, $[S_1]$ and $[T_1]$ were obtained by minimizing the set of differential equations (20) and (21) with $[n]$ given by equation (7), using the python package scipy[161,162].

$$\frac{d}{dt}[S_1] = -\underbrace{(k_r^S + k_{nr}^S + k_{ISC})}_{=k^S}[S_1] + k_{RISC}[T_1] + \frac{1}{4}\gamma_{TT}[T_1]^2 \tag{20}$$

$$-\gamma_{ST}[T_1][S_1] + \frac{1}{4}\gamma[n]^2$$

$$\frac{d}{dt}[T_1] = -\underbrace{(k_r^T + k_{nr}^T + k_{RISC})}_{=k^T}[T_1] + k_{ISC}[S_1] - \frac{5}{4}\gamma_{TT}[T_1]^2 + \frac{3}{4}\gamma[n]^2 \tag{21}$$

## Calculation of target value

The optical power flux $\Phi$ leaving an OLED relates to the current density $J$ as

$$\Phi_e = \int \Phi_e(\lambda)d\lambda = \int \frac{hc}{\lambda}I(\lambda)\eta_{EQE}\frac{J}{e}d\lambda = \frac{hc}{e}\eta_{EQE}J\int\frac{1}{\lambda}I(\lambda)d\lambda \tag{22}$$

where $I(\lambda)$ is the relative intensity of the OLED at wavelength $\lambda$, $\eta_{EQE}$ is the ratio of photons leaving the OLED to the number of electrons flowing around the electrical circuit (EQE).

The total luminous flux $\Phi_V$ can be calculated from the optical power flux using the photonic sensitivity curve $V(\lambda)$ as

$$\Phi_V = K_m\int\Phi_e(\lambda)V(\lambda)d\lambda = K_m\frac{hc}{e}\eta_{EQE}J\int\frac{1}{\lambda}I(\lambda)V(\lambda)d\lambda \tag{23}$$

where $K_m = 683\ lm\ W^{-1}$ is a fudge factor called the peak response.

Under the assumption of Lambertian emission, the luminance $L_V$ of the OLED is then given as follows.

$$L_V = \frac{\Phi_V}{\pi} = K_m\frac{hc}{\pi e}\eta_{EQE}J\int\frac{1}{\lambda}I(\lambda)V(\lambda)d\lambda \tag{24}$$

Therefore, the current density required to generate a given luminance by an OLED with a given normalized spectrum and given EQE is given as follows.

$$J = \frac{1}{K_m}\frac{\pi e}{hc}\frac{L_V}{\eta_{EQE}}\left[\int\frac{1}{\lambda}I(\lambda)V(\lambda)d\lambda\right]^{-1} \tag{25}$$

For the calculation of the target value, we have taken three assumed spectra for red, green and blue with a Gaussian shape and a full-width at half-maximum of 45, 30 and 15 nm, respectively. The centre wavelength was selected so that the colours of the three spectra are as close as possible to the primary colours of the BT.2020 standard in the CIE 1931 colour space, which are given by the coordinates (0.708, 0.292), (0.170, 0.797) and (0.131, 0.046), respectively[46].

The calculation was performed at an EQE of 22.5% for $L_V = 1,000\ cd\ m^{-2}$, indicating a maximum EQE of 25%. The correlation for $J_{90}$ to the FOM is

taken from the simulated relationship shown in Extended Data Fig. 4a for $\gamma_{TT} = 10^{-13}\ cm^3\ s^{-1}$ and $\gamma_{ST} = \gamma_{TP} = 0$ as follows.

$$\log\left(\frac{J_{90}}{1\ Am^{-2}}\right) = 2\log\left(\frac{k_r^S K_{eq}}{1\ s^{-1}}\right) - 8.3764 \tag{26}$$

$$\Leftrightarrow k_r^S K_{eq} = 10^{\frac{\log\left(\frac{J_{90}}{1\ Am^{-2}}\right)+8.3764}{2}}\ s^{-1} \tag{27}$$

## Data availability

The data supporting this publication can be accessed at https://doi.org/10.17630/d1439596-7eef-44ae-90cd-667b70588896.

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

**Acknowledgements** We are grateful to the Engineering and Physical Sciences Research Council of the UK for financial support through grant nos. EP/R035164/1 and EP/P010482/1. We are grateful to K. Yoshida for discussions relating to the target values for the FOM.

**Author contributions** I.D.W.S. proposed an FOM approach to efficiency roll-off and wrote the first draft of the paper. L.Z. performed the data acquisition. S.D. provided the derivation. L.Z., S.D. and I.D.W.S. performed the analysis. All authors reviewed and edited the manuscript.

**Competing interests** The authors declare no competing interests.

**Additional information**
**Correspondence and requests for materials** should be addressed to E. Zysman-Colman or I. D. W. Samuel.

(a)

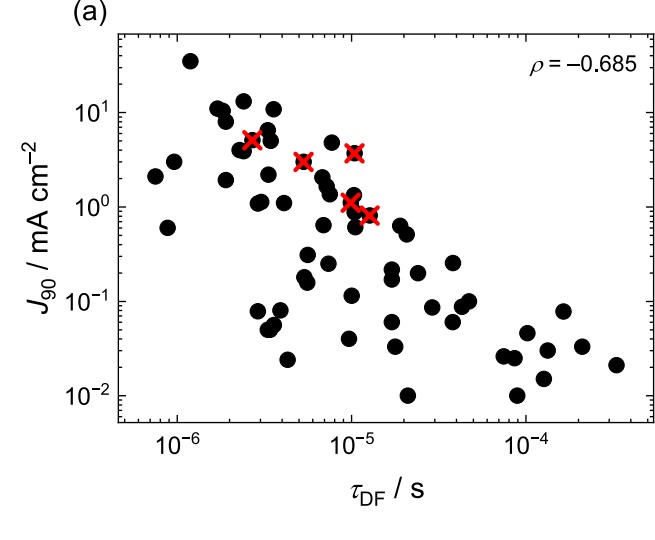

(b)

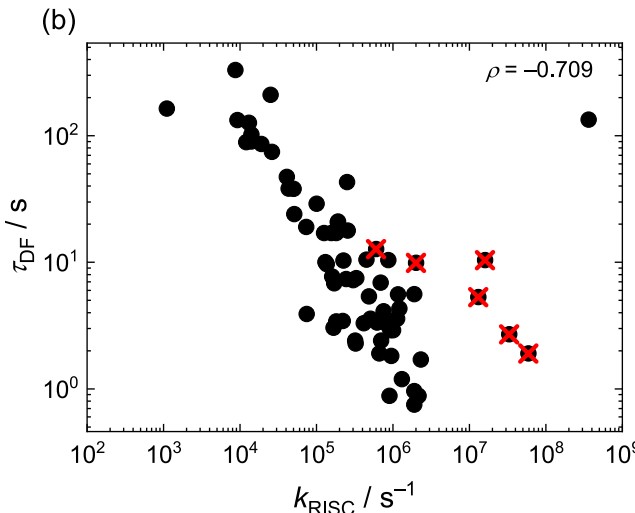

(c)

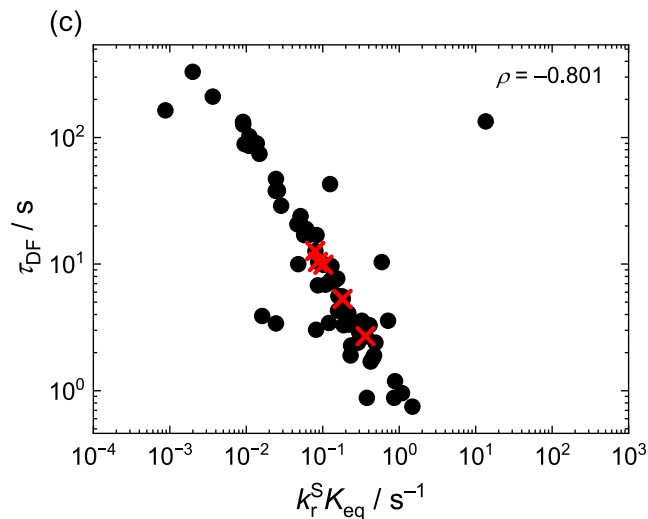

**Extended Data Fig. 1 | Correlation with delayed fluorescence lifetime $\tau_{DF}$.**
(a) Dependence of $J_{90}$ on $\tau_{DF}$ with a Spearman correlation, $\rho$, of −0.685.
(b) Dependence of $\tau_{DF}$ on $k_{RISC}$ ($\rho = -0.709$). (c) Dependence of $\tau_{DF}$ on $k_r^S K_{eq}$
($\rho = -0.801$). Red crosses identify TADF molecules containing heavy atoms that enhance SOC, which leads to an increase in $k_{RISC}$.

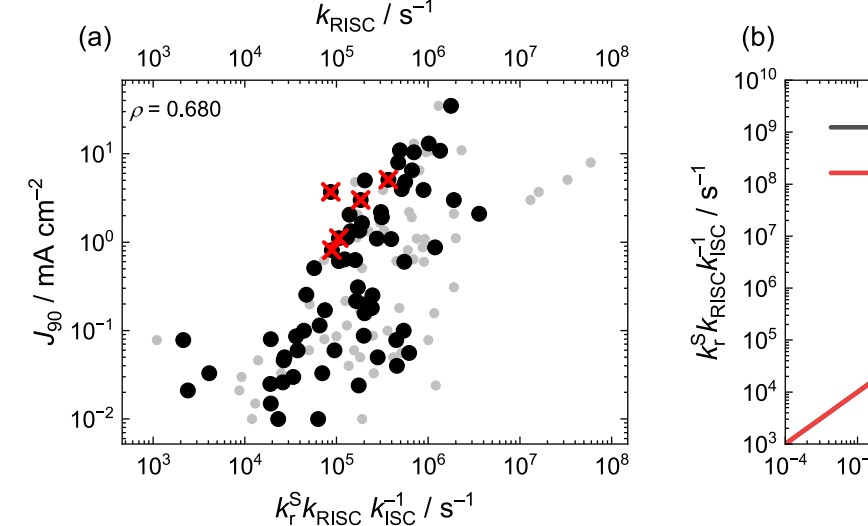

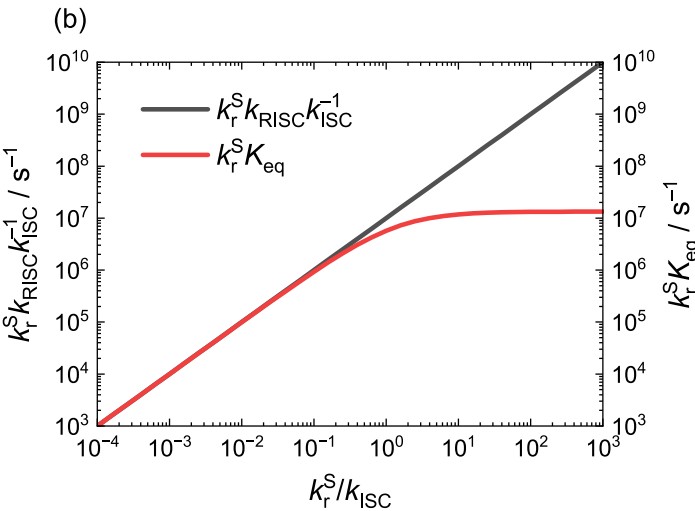

**Extended Data Fig. 2 | Simplified Figure of Merit.** (a) Correlation between $J_{90}$ and the simplified FOM of $k_r^S k_{RISC}/k_{ISC}$ with a Spearman correlation of $\rho = 0.680$, showing a better correlation than $k_{RISC}$ but a less precise predictor than the FOM of $k_r^S K_{eq}$. Red crosses identify TADF molecules containing heavy atoms that enhance SOC, which leads to an increase in $k_{RISC}$. In grey circles, the correlation of $J_{90}$ and $k_{RISC}$ from Fig. 3 is displayed for comparison. (b) Deviation between the FOM of $k_r^S K_{eq}$ and its simplification of $k_r^S k_{RISC}/k_{ISC}$ for $k_{RISC} = 10^7\,s^{-1}$ showing a deviation between the FOMs for systems with competitive $k_r^S$ and $k_{ISC}$.

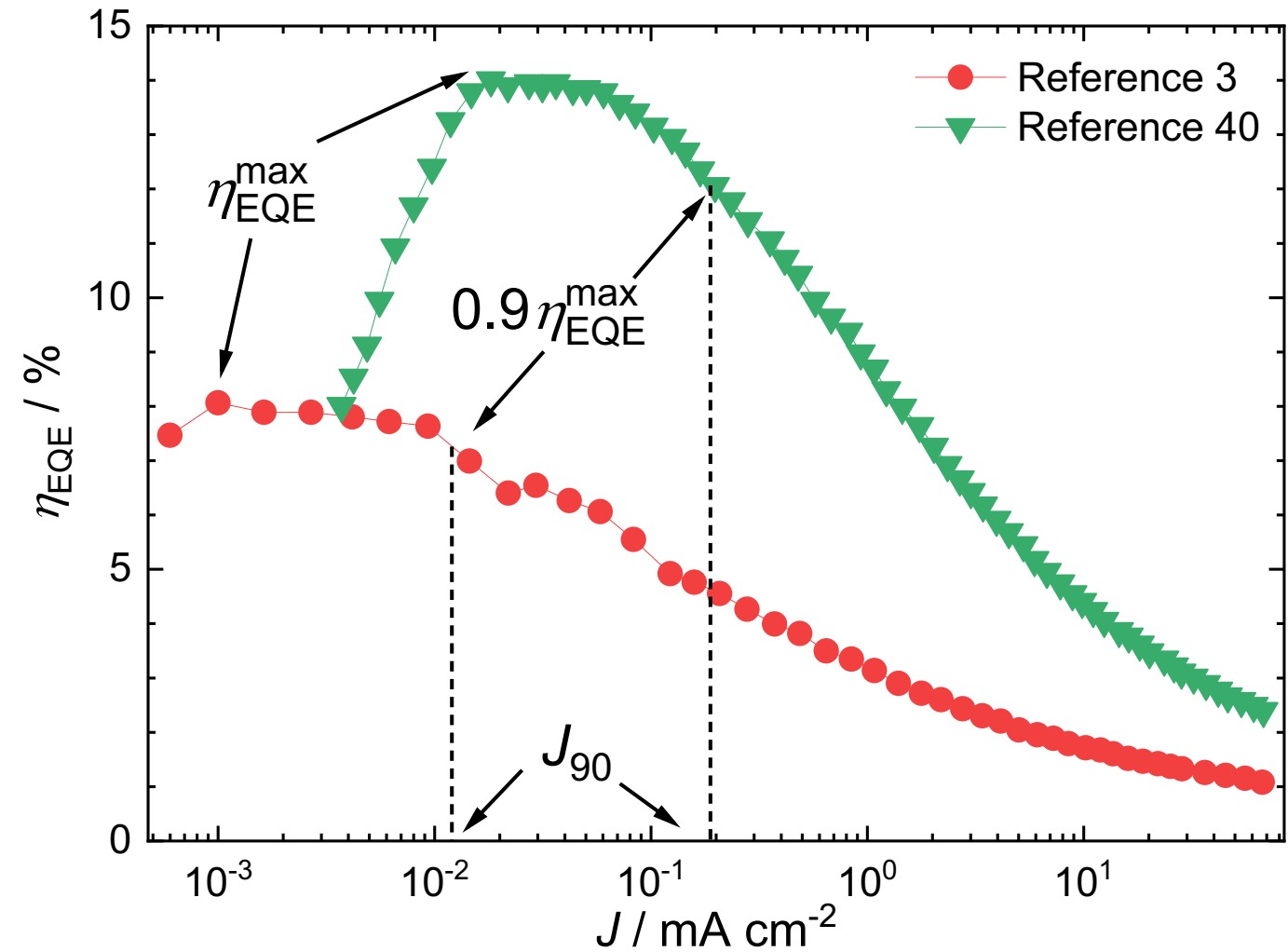

**Extended Data Fig. 3 | Device influence on Roll-off.** Comparison of efficiency roll-off of two literature 2CzPN OLEDs showing different $J_{90}$ because of different efficiency rise at low current densities[3,40].

(a)

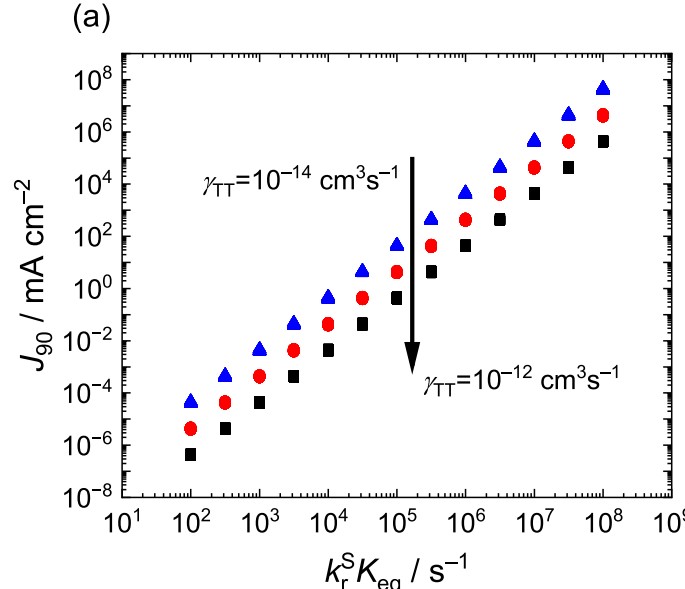

(b)

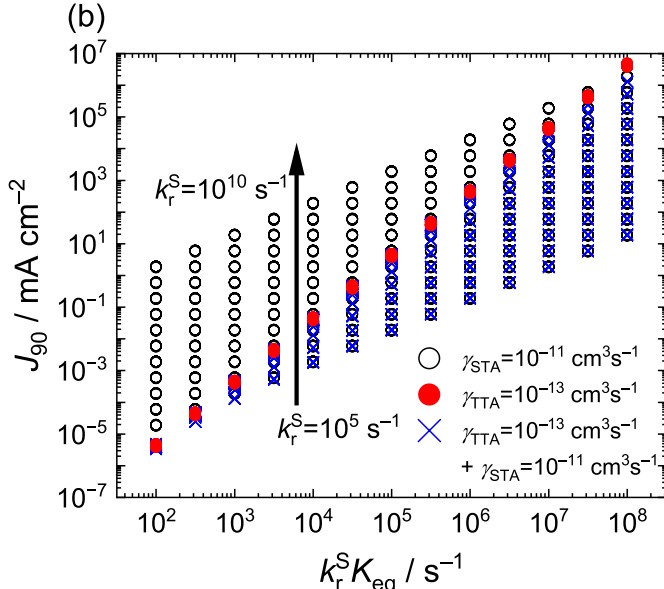

**Extended Data Fig. 4 | Impact of STA and TTA on roll-off.** The impact of STA and TTA on the correlation between $J_{90}$ and $k_r^S K_{eq}$ calculated for a simplified three-level system with $k_r^S$ between $10^5$ s$^{-1}$ and $10^{10}$ s$^{-1}$, $k_{ISC}/k_r^S$ between $10^{-1}$ and $10^3$ and $k_r^S K_{eq}$ between $10^2$ s$^{-1}$ and $10^8$ s$^{-1}$ (a) for three different TTA rate constants and (b) for a particular STA rate, a particular TTA rate and a particular combination of STA and TTA rate.

**Extended Data Table 1 | Parameters for Extended Data Fig. 4**

| $k_r^S \; (\mathrm{s}^{-1})$ | $\frac{k_{\mathrm{ISC}}}{k_r^S}$ | $k_r^S K_{\mathrm{eq}} \; (\mathrm{s}^{-1})$ |
|---|---|---|
| $10^{5.0}$ | $10^{-1.0}$ | $10^{2.0}$ |
| $10^{5.5}$ | $10^{-0.5}$ | $10^{2.5}$ |
| $10^{6.0}$ | $10^{0.0}$ | $10^{3.0}$ |
| $10^{6.5}$ | $10^{0.5}$ | $10^{3.5}$ |
| $10^{7.0}$ | $10^{1.0}$ | $10^{4.0}$ |
| $10^{7.5}$ | $10^{1.5}$ | $10^{4.5}$ |
| $10^{8.0}$ | $10^{2.0}$ | $10^{5.0}$ |
| $10^{8.5}$ | $10^{2.5}$ | $10^{5.5}$ |
| $10^{9.0}$ | $10^{3.0}$ | $10^{6.0}$ |
| $10^{9.5}$ | | $10^{6.5}$ |
| $10^{10}$ | | $10^{7.0}$ |
| | | $10^{7.5}$ |
| | | $10^{8.0}$ |

Generating parameters for the tested set of theoretical emitters shown in Extended Data Fig. 4.