## [Peer Review File · Nature]

Manuscript Title: A Figure of Merit for Efficiency Roll-off in TADF-based Organic LEDs

Reviewer Comments & Author Rebuttals

Reviewer Reports on the Initial Version:

Referees' comments:

Referee #1 (Remarks to the Author):

In recent years, OLEDs have made remarkable progress, in terms of both basic research and practical applications. Among them, TADF occupies a major role, and the efficiency roll-off problem is one of the most important issues. Its solution is the concern of all researchers in this field, so it is worth considering and its publication is timely.

1) As the authors describe, attempts have been made to increase kRISC to solve the efficiency roll-off problem. This is because RISC is inherently forbidden and has remained a rate-limiting process in TADF that made the RISC possible. Typically, kRISC was more than two orders of magnitude smaller than krS. However, recent molecular design has solved this problem. That is, molecules with kRISC larger than krS are beginning to be obtained. As a result, it has become clear that increasing kRISC alone is not enough to further improve efficiency roll-off, and other processes need to be considered. Researchers in this field now believe that in addition to achieving large krS and large kRISC simultaneously, a kISC smaller than krS is important to avoid exciton return to T1. Please clarify that this study is more than the above relationship ($krS > (or \sim) kISC \sim kRISC$) shared by researchers in this field.

2) I would like to confirm that J90 is an appropriate indicator.

The authors use J90 as a measure of efficiency roll-off. This seems to be a direct indicator, but as the authors point out in Extended Data Fig. 2, the maximum value of J varies, and the J90 itself varies by about one order of magnitude.

3) Both FOM1 and FOM2 proposed by the authors consist of parameters of TADF molecules, krS, kISC, and kRISC. Is the FOM in this study related to the design of TADF molecules? If so, it looks fine. On the other hand, J90 varies greatly depending on other factors. For example, it varies greatly with the doping concentration of the TADF molecule in the device. Perhaps, the authors need to consider excluding such effects from the J90 index.

4) Fig. 4a and b appear to give similar correlations. Both are similar functions of krS, kISC, and kRISC, so the results are reasonable (FOM2 becomes FOM1 when $krS \ll kISC$). Fig. 3b also appears to show a good correlation in the region of $kRISC > 10^7$, although the number of data is small. Fig. 3c does not show a good correlation. What could be the reason for this? Fig. 4a seems to be essentially the same as correlating with dEst, because $kRISC/kISC$ cancels out the effect of SOC by taking the ratio. Is the key

point that it is a function of exponential or that krS is multiplied? Related to this, since $kRISC$ considers SOC in addition to $dEst$, the comparison in Fig. 3b, c illustrates the importance of SOC. It might be a good idea to introduce SOC into the FOM.

5) Recently, RISCs via T2 have been found frequently. How do the authors consider this case? (Equations (S1) and (S2) do not deal with T2.)

6) It looks both conventional TADF and MR-TADF are analyzed. Is it possible to treat them with the same concept?

7) Please specify the target area. For example, in Figures. Or as FOM values.

8) Fig. 1c is an important summary. Fig. 3d is suggestive that $kRISC/kISC$ shows constant values below $\Delta Est = 0.1$ eV.

9) Line 397: in equation (4), $krS/kISC \rightarrow kISC/krS$

10) Line 866: Fig. 4b \rightarrow Fig. 3b

Referee #2 (Remarks to the Author):

The presented manuscript analyses very important and widely accessed topic of roll-off in OLED devices, limiting its practical applicability. A number of TADF compounds with the corresponding OLED devices were studied seeking to find a dominating parameters, describing the potential extent of roll-off in OLED device. Authors developed a model providing relation of roll-off parameter $J90$ with relaxation parameters of singlet and triplet population in TADF materials.

Authors correlate the proposed Figures of merit (FOMS) to the values obtained for 66 devices published by other authors. The results however are widely spread.

- Three correlations are discussed, namely $J90$ with $kRISC$, FOM1 and FOM2, described by correlation coefficients. Since the sample size is statistically not very large and strongly spread, the obtained Spearman correlation coefficients of 0.638, 0.680 and 0.700 looks very similar and it is very difficult to observe any improvement of FOM2 as compared to simple $J90 \sim kRISC$ comparison. In order to prove the excellence of FOM2 over other parameters, broader statistical analysis should be performed, indicating that there is a statistical significance in the $J90$ dependences with all three other parameters. Moreover, the spread of data should also be described by statistical parameters.

- The impact of material degradation is not properly addressed in the study. That brings very large concern on the validity of $J90$ values used.

- The main and totally undiscussed issue is conformational disorder in the solid-state. As it was shown previously, solid surrounding leads to conformational distribution of TADF molecules, enabling the dispersion of singlet-triplet energy gaps. Typical TADF compound may have singlet-triplet energy gap distribution spanning for nearly 100 meV, while the spectroscopically assessed value is just an average of all gaps (see 10.1021/acs.jpcc.9b08269). This leads to remarkable variation of rISC values for the same TADF compound, spanning over several orders of magnitude (see 10.1021/acs.jpcllett.2c01864). As the delayed fluorescence decay in films usually is strongly multiexponential, only very rough fitting can be performed, resulting in the average DF decay time, as well as only rough average of rISC rate. Moreover, weak DF decay usually lasts up to ms time-range, making it very difficult to assess it spectroscopically (see 10.1039/d3tc00482a, 10.1103/PhysRevB.68.075208). Usually, only simple TSCPC or streak cameras are used, assessing only the initial DF decay. In such case, large errors (e.g. order of magnitude) in estimation of rISC rates occurs (see 10.1021/acs.jpca.0c10391).

- P.34 Extended Data Fig. 1 a, c, and d - errors in delayed fluorescence lifetime measures and scales.

Summarizing, though the study is important, however it is based on very diverse data with hardly ensured validity. And above all, no statistical proof of superiority of FOM2 was given. This manuscript cannot be published in Nature.

Referee #3 (Remarks to the Author):

Through summarizing and analyzing previously reported data of TADF-OLED devices and photo-physics, Samuel, Zysman-Colman, and co-workers proposed figures of merit (FOMs) for efficiency roll-off, and discussed correlations between FOMs and J90. Although these FOMs could be useful for TADF-OLED community, obviously it is too specific to meet broad readership of Nature. Here, I also have some comments and questions that need to be addressed by authors:

(1) the number of the used data is small. Thus, the reliability of their conclusions could not be guaranteed. And, all used data in this paper should be listed in SI.

(2) rates of RISC and ISC cannot be measured directly, but rather fitted. The spread of data points in Fig. 4 could be induced by inaccurate fitting of kRISC and kISC.

(3) I understand the point made by authors. If we directly use delayed lifetime as FOM including both effects of RISC and S1-state radiative decay, what happens? In fact, delayed lifetime has been used as a good descriptor to screen TADF emitters, see literature Nature Mater 15, 1120–1127 (2016).

(4) in Figure 1, it is unfair to compare EQE values of OLEDs emitting different colors at the same luminance (1000 nits), as the same luminance may correspond to different photon counts.

Author Rebuttals to Initial Comments:

A Figure of Merit for Efficiency Roll-off in Thermally Activated Delayed Fluorescent Organic Light-Emitting Diodes

Response to Referees' Comments

We are grateful to have the opportunity to respond to the referees' comments. From their comments we can see that efficiency roll-off is an important issue and there is a range of opinions on what the key factors affecting it are. We also learn that some aspects of our idea need to be expressed more clearly, so explain some context before addressing the specific points.

Ultimately physical principles set limits on what can be achieved. For example, we cannot make single layer organic light-emitting diodes with quantum efficiency over 100%, and we can only achieve 33% power conversion efficiency for a single junction solar cell. It is possible to make such statements without knowing every detail of every material and device made. In our manuscript we are seeking to highlight the combination of intrinsic material properties that ultimately limits efficiency roll-off, which is a crucial device property for the large worldwide OLED community – both academic and industrial.

On the basis of a wide range of collected experimental data, we show unambiguously that efficiency roll-off is a serious problem for thermally activated delayed fluorescent organic light emitting diodes (TADF-OLEDs). As in other OLEDs it is believed to be primarily due to interactions with triplets and so to reduce efficiency roll-off, the triplet population needs to be reduced, which in many cases amounts to reducing the triplet lifetime. At present most of the literature and TADF community are focussed on increasing the rate of reverse intersystem crossing, k_{RISC} to address this problem (specific comments of the referees are dealt with below), which however does not always work or in some cases lead to opposite effects. The key idea of our *Analysis* is that because TADF involves a dynamic equilibrium between singlets and triplets, a combination of rate constants controls the triplet population (and delayed fluorescence lifetime). This is simply a matter of the physics (or physical chemistry) of the emission process, and the fact that it has not yet been clearly stated in this way is an important gap that we are seeking to fill in order to guide future materials design.

A particular feature of the article is the formulation of figures of merit (FOM) to guide the field. This is inspired by my experience of the field of nonlinear optics around 30 years ago. At that time some people focussed on increasing nonlinearity, others warned absorption needed to be considered, still others that two-photon absorption was the main problem. People debated which was most important until George Stegeman produced a figure of merit for optical switching which, in time, transformed the field by clearly showing what combination of parameters was needed to achieve a particular goal. The field of TADF today is similar – there are various parameters to consider, and people think they understand the general principles, yet they haven't so far combined those insights into a clear figure of merit that can guide the field. At present a synthetic chemist seeking to improve efficiency roll off would see the emphasis on increasing the rate of reverse intersystem crossing (RISC) in the literature and would therefore try to increase spin-orbit coupling. The proposed figure of merit would enable them to see quickly that in fact increasing spin-orbit coupling would also

increase the forward process of intersystem crossing (ISC) and that a better strategy would be to increase the radiative rate constant for the singlet. In other words, a large RISC alone does not necessarily guarantee a small roll-off. The TADF community needs to reconsider molecular design principles to achieve both high efficiency and low efficiency roll-off.

We firmly believe that the clear statement of the relevant combination of parameters arising from solid physical principles can steer the field and provide it with both the stimulus and route to solving the severe problem of efficiency roll-off. It is important to explain that the figure of merit is really describing the best that could be achieved with a material. Many factors could lead to worse performance, but the best that can be achieved is still a powerful guide for the field.

We acknowledge that in the previous version of the paper it may not have been clear that the figure of merit at the core of the paper comes from a physical argument to guide materials design. It is possible that because we assessed correlations with many potential parameters that it appeared we just took the best of many combinations, whereas in fact we were guided by considering of the dynamic equilibrium at the heart of TADF. Accordingly, we have revised the manuscript extensively and removed some of the unnecessary correlations.

We understand that two key concerns about the submitted manuscript were:

1. Is the proposed figure of merit really different from existing guides in the literature?

Here, we first outline the existing guides, and then the differences (and advantages) of the proposed Figure of Merit (FOM).

a) *The existing guides*

Generally speaking, there are two kinds of guides to mitigate the efficiency roll-off, i.e., molecular design guides and device optimization guides. The former is our focus here, because it defines the upper/best limits that a device can achieve.

For TADF molecular design guides, most of the literature emphasises k_{RISC} as the key parameter and there have even been conferences devoted to this topic. The referees have a more sophisticated approach, though the first and third referees suggest very different alternative guides for addressing efficiency roll-off – showing the kind of fragmentation in the field that means there is a need for our article and could be unified by a FOM approach. The first referee suggests $k_{IS} > \sim k_{ISC} \sim k_{RISC}$, whereas the third referee suggests that delayed fluorescence lifetime should be minimised.

In terms of device optimisation, current understanding is that charge capture should be balanced, and the recombination zone should be as broad as possible to reduce the triplet population density. This strategy is applicable to any material, but after applying it, the efficiency roll-off is ultimately limited by the properties of the emitter layer as captured by our FOM. And it is important to keep in mind that usually the entire field of organic optoelectronics is limited by materials performance, rather than device design.

b) *Differences (and Advantages) of the proposed FOM*

Our FOM is different because it gives the precise combination of the rate constants that should be targeted and gives a quantitative measure of the expected benefit. Furthermore, it clearly shows the trade-off between any of these rate constants. **More importantly, it really works!** Here, we can give an example of how our FOM can be applied to understand the literature. It is just one of many other similar cases and nicely demonstrates that our work is resonating with many other researchers, who have already noticed the “ k_{RISC} ” issue.

In “Donor Extension on Spiro-Acridine Enables Highly Efficient TADF-OLEDs with Relieved Efficiency Roll-Off.” Adv. Funct. Mater. 2023, **33**, 2211696.

<https://doi.org/10.1002/adfm.202211696>, three TADF molecules were synthesized. Using the same device structure, the authors tried to relate photophysical properties to the device efficiency roll-off, which however turned out to be a failure. Two tables are from the paper.

Table 3. Data of transient PL decay of 2S-TRZ, IT-TRZ, and IA-TRZ in doped films (10 wt% in PPF).

Emitter	τ_p [ns]	τ_d [μ s]	Φ_{PL} [%]	R_p [%]	R_d [%]	k_r [10^7 s^{-1}]	k_{ISC} [10^7 s^{-1}]	k_{RISC} [10^5 s^{-1}]	$k_{\text{ISC}}/k_{\text{RISC}}$
2S-TRZ	26	2.4	99	46	54	1.74	2.06	8.84	23.3
IT-TRZ	28	2.2	98	45	55	1.57	1.94	9.71	19.9
IA-TRZ	28	2.4	96	36	64	1.24	2.32	11.4	20.4
D2T-TRZ ^[36]	26	2.2	97	34	63	1.39	2.60	13.1	19.8
p-TRZ ^[38]	17	2.4	99	55	44	3.28	2.60	7.40	35.1

Table 4. Data of OLEDs based on 2S-TRZ, IT-TRZ, and IA-TRZ as emitters.

Emitting layer	V_{on} [V]	L_{max} [cd m^{-2}]	CE^{a} [cd A^{-1}]	EQE^{a} [%]	Efficiency ^b roll-off [%]
2S-TRZ (10 wt%)	3.6	42 055	95.0 / 81.7 / 59.6	32.6 / 28.6 / 21.1	12/35
IT-TRZ (10 wt%)	3.6	51 590	105.5 / 89.8 / 70.7	35.8 / 30.9 / 24.7	14/31
IA-TRZ (10 wt%)	3.6	36 508	94.1 / 80.9 / 56.5	32.0 / 27.9 / 20.1	13/37
2S-TRZ (20 wt%)	3.2	76 964	108.6 / 104.1 / 87.8	35.6 / 34.3 / 29.3	4/18
IT-TRZ (20 wt%)	3.2	105 907	111.6 / 110.7 / 98.7	36.1 / 36.0 / 32.3	0.1/11
IA-TRZ (20 wt%)	3.3	39 557	90.4 / 78.2 / 57.8	29.7 / 26.3 / 19.9	11/33

V_{on} , L_{max} refers to turn-on voltage recorded at 1 cd m^{-2} and maximum luminance. ^acurrent efficiency and EQE of maximum value/under 100/1000 cd m^{-2} ; ^befficiency roll-off under 100 and 1000 cd m^{-2} .

As we can see if only k_{RISC} is considered as the authors and many other researchers usually do, **IA-TRZ** should have the lowest roll-off. If only k_r is considered, **2S-TRZ** should have the lowest roll-off. However, it is **IT-TRZ** that showed the lowest roll-off (at 1000 cd/m^2). The authors also tried $k_{\text{ISC}}/k_{\text{RISC}}$, but it is not the right indicator. If we use our FOM (last column of the table below) to analyse the photophysical data, the best roll-off performance should be the **IT-TRZ** device because it has the largest FOM of $4.9 \times 10^5 \text{ s}^{-1}$ (see table below). Our FOM not only works, but can predict.

	k_r [10^7 s^{-1}]	k_{ISC} [10^7 s^{-1}]	k_{RISC} [10^5 s^{-1}]	$k_r k_{\text{ISC}}$ [10^5 s^{-1}]
2S-TRZ	1.74	2.06	8.84	4.6
IT-TRZ	1.57	1.94	9.71	4.9
IA-TRZ	1.24	2.32	11.4	4.3

Last but not least, we respond about delayed fluorescence as a possible guide here. It is already in the manuscript as extended data Figure 1(a). As the referee notes, it gives a good

correlation and is suitable for screening materials that have been made. But it does not in itself guide the design of materials with shorter delayed fluorescence lifetime. Most researchers understand delayed fluorescence lifetime should be shortened, but believe this should be achieved by increasing the rate of RISC, whereas in fact consideration of the physics of the situation means our FOM is the quantity to focus on increasing – for example by increasing oscillator strength to increase k_r^S .

We also notice that our FOM has a stronger correlation with τ_{DF} than that of k_{RISC} with τ_{DF} as shown in Extended data Figs. 1(b) and (c) and the figure below, i.e., the dependence of τ_{DF} on k_{RISC} has much greater spread than its dependence on FOM. This shows that our FOM is a more reliable indicator similar to that of τ_{DF} but as noted above gives more physical insight into the processes controlling the triplet population, and hence is more suitable for guiding molecular design.

Fig.R1 Dependence of τ_{DF} on (left) k_{RISC} , and (right) FOM. Red lines are guide-to-eyes. Red crosses identify TADF molecules containing heavy atoms that enhance SOC, which leads to an increase in k_{RISC} .

2. Does the scatter in the plots of efficiency roll-off (J_{90}) vs FOM suggest that the FOM is in fact not well chosen?

Efficiency roll-off results both from the intrinsic properties of the emitter and the extrinsic properties of the device (such as charge balance and recombination profile). Trying to optimise everything at once is difficult, so a clear guide to optimise materials will be very useful.

In particular our FOM arises from considering the physics of the emission process to reduce triplet population. Essentially it describes the best that can be achieved with a material. As devices are complicated, many factors can mean that actual devices fall below this best case. But in fact, this makes such a FOM all the more valuable because otherwise the enormous number of combinations of multiple materials parameters and multiple device parameters

makes it very difficult to establish guiding principles and also leads to material development being slow, tedious and expensive.

In the OLED field a key early breakthrough was developing measurements of the photoluminescence efficiency of the light-emitting materials. This gave the maximum potentially achievable efficiency. Although devices at the time were far below it, and variable, it nevertheless greatly helped the development of materials.

We next consider the specific points raised. We put the comments in blue and our response in black.

Referees' comments:

Referee #1 (Remarks to the Author):

In recent years, OLEDs have made remarkable progress, in terms of both basic research and practical applications. Among them, TADF occupies a major role, and the efficiency roll-off problem is one of the most important issues. Its solution is the concern of all researchers in this field, so it is worth considering and its publication is timely.

1) As the authors describe, attempts have been made to increase k_{RISC} to solve the efficiency roll-off problem. This is because RISC is inherently forbidden and has remained a rate-limiting process in TADF that made the RISC possible. Typically, k_{RISC} was more than two orders of magnitude smaller than k_{rS} . However, recent molecular design has solved this problem. That is, molecules with k_{RISC} larger than k_{rS} are beginning to be obtained. As a result, it has become clear that increasing k_{RISC} alone is not enough to further improve efficiency roll-off, and other processes need to be considered. Researchers in this field now believe that in addition to achieving large k_{rS} and large k_{RISC} simultaneously, a k_{ISC} smaller than k_{rS} is important to avoid exciton return to T1. Please clarify that this study is more than the above relationship ($k_{\text{rS}} > (\text{or } \sim) k_{\text{ISC}} \sim k_{\text{RISC}}$) shared by researchers in this field.

We are pleased the referee agrees that efficiency roll-off is “one of the most important issues” for the TADF field. As mentioned above, our FOM gives the precise combination of rate constants that should be targeted and gives a quantitative measure of the expected benefit. Furthermore, it clearly shows the trade-off between any of these rate constants. Actually, the scenario the referee suggests is not common (see Tables 3 and 4 of the Advanced Functional Materials paper discussed above and also the collected data in Fig. 4).

Nevertheless, our FOM is very useful for understanding the regime of $k_{\text{r}}^{\text{S}} > k_{\text{ISC}}$. FOM can be written as $k_{\text{r}}^{\text{S}} K_{\text{eq}}$ and indicates that balance between these rates must be achieved. When $k_{\text{r}}^{\text{S}}/k_{\text{ISC}}$ exceeds 0.1, the FOM shows a deviation from the trend of the ratio of $k_{\text{r}}^{\text{S}}/k_{\text{ISC}}$ as shown in the plot below. This has some interesting consequences for hyperfluorescent devices (i.e. devices using a TADF molecule as the sensitizer followed by energy transfer to terminal emitter), as any outcompeting of ISC by FRET of more than 2 orders of magnitude does not

reduce the relative T₁ population on the TADF molecule. Thus, giving a concentration limit of the terminal emitter above which the roll-off cannot be improved.

We have modified the manuscript to include the figure below in Extended data Fig 2(b), and have also focussed on a single FOM (FOM2 of the previously submitted version) because it works in the regime of radiative decay faster than ISC that the referee refers to, as well as the more common regime where radiative decay is slower than RISC.

Figure showing how FOM/ k_{RISC} depends on k_r^S/k_{ISC} .

2) I would like to confirm that J90 is an appropriate indicator.

The authors use J90 as a measure of efficiency roll-off. This seems to be a direct indicator, but as the authors point out in Extended Data Fig. 2, the maximum value of J varies, and the J90 itself varies by about one order of magnitude.

We are happy the referee agrees that this is an appropriate measure of efficiency roll-off.

3) Both FOM1 and FOM2 proposed by the authors consist of parameters of TADF molecules, k_r^S , k_{ISC} , and k_{RISC} . Is the FOM in this study related to the design of TADF molecules? If so, it looks fine. On the other hand, J90 varies greatly depending on other factors. For example, it varies greatly with the doping concentration of the TADF molecule in the device. Perhaps, the authors need to consider excluding such effects from the J90 index.

Yes, the parameters in the FOM are related to the design of TADF molecules. For example, k_{ISC}/k_{RISC} depends on the energy difference between singlet and triplet. To date that has been

minimised by reducing overlap of donor and acceptor orbitals. This in turn reduces k_rS . Researchers are aware they must not reduce k_rS too much, but our FOM quantifies the trade-off between these parameters and so informs molecular design (for example k_rS can be calculated in a fairly simple way; k_{RISC} and k_{ISC} can be calculated in a more advanced calculation).

We agree that J_{90} is affected by other factors such as doping concentration and charge balance, but as explained earlier, a guide for materials properties is still very useful – just like photoluminescence quantum yield measurements helped the OLED field. Very likely researchers will optimise device factors empirically as far as possible, arriving at a device limited by the intrinsic properties of the material. Our FOM indicates the desired combination of material properties.

4) Fig. 4a and b appear to give similar correlations. Both are similar functions of k_rS , k_{ISC} , and k_{RISC} , so the results are reasonable (FOM2 becomes FOM1 when $k_rS \ll k_{ISC}$). Fig. 3b also appears to show a good correlation in the region of $k_{RISC} > 10^7$, although the number of data is small. Fig. 3c does not show a good correlation. What could be the reason for this?

Fig 3c was plotted because ΔE_{ST} is one of the key parameters pursued in the field, so the point of the figure was to show there was not a good correlation so that readers are aware that on its own it does not describe efficiency roll-off. The reason for this is that although it affects k_{RISC}/k_{ISC} , the rate of radiative decay is also a key parameter for obtaining light emission and shortening delayed fluorescence lifetime. We have removed this plot from the revised version because (as explained above) it contributes to the impression that we surveyed every possible predictor of efficiency roll-off and selected one which happened to work, whereas we actually developed the FOM based on consideration of triplet population and then tested it.

Fig. 4a seems to be essentially the same as correlating with dE_{ST} , because k_{RISC}/k_{ISC} cancels out the effect of SOC by taking the ratio. Is the key point that it is a function of exponential or that k_rS is multiplied?

The key point is indeed the inclusion of k_rS . This removes useful “product” (light emission) from the equilibrium.

Related to this, since k_{RISC} considers SOC in addition to dE_{ST} , the comparison in Fig. 3b, c illustrates the importance of SOC. It might be a good idea to introduce SOC into the FOM.

The importance of spin-orbit coupling is that it facilitates RISC and ISC. However, when we consider the equilibrium, it is the relative rate of these processes which matters, not the absolute value of either one. This has a very important implication for molecular design which is that increasing spin-orbit coupling alone may not help – unless the relative rate of RISC and ISC can be changed. There is experimental evidence that increasing spin-orbit coupling is not as helpful as expected (references 22 and 23 in the manuscript).

5) Recently, RISCs via T2 have been found frequently. How do the authors consider this case? (Equations (S1) and (S2) do not deal with T2.)

The key point for efficiency roll-off is the rate of RISC relative to ISC. It does not matter whether intermediate states are involved.

Furthermore, the inter-conversion between the two triplet states has been reported to be fast in comparison to other rates involved in TADF. Thus, to analyse the kinetics of the excited states involved TADF, the triplet states can be viewed as one. The lesson of our analysis can in any case be generalised to say that to reduce the EQE roll-off a material should be selected which intrinsically leads to a reduced exciton population in the triplet manifold relative to the singlet population. This can be achieved by maximising our figure of merit.

6) It looks both conventional TADFs and MR-TADFs are analyzed. Is it possible to treat them with the same concept?

Yes, because efficiency roll-off in both cases is due to triplets and mitigated by reducing the triplet population. Furthermore, the same concept also treats materials incorporating heavy atoms to increase k_{RISC} .

7) Please specify the target area. For example, in Figures. Or as FOM values.

The target will depend on the intended use and display design, but we are happy to provide examples of target values, and have added text to explain this on page 11 of the manuscript

“Target values of the FOM will depend on the requirements of particular applications, as well as device design and severity of bimolecular effects. We estimate values of FOM required for materials with chromaticity close to the BT2020 standard,⁴⁵ and with Gaussian emission spectra of width 15 nm in the blue, 30 nm in the green and 45 nm in the red. We use the calculation for Extended Data Figure 4a with $\gamma_{\text{TT}} = 10^{-13} \text{ cm}^3 \text{ s}^{-1}$ and find the FOM required to achieve 90% of a peak EQE of 25% at a brightness of 1000 cd m^{-2} . We find that for a deep blue emitter ($\lambda_{\text{max}} = 467 \text{ nm}$, CIE (0.131, 0.049)) a FOM of at least $1.5 \times 10^5 \text{ s}^{-1}$ is required. For a green emitter ($\lambda_{\text{max}} = 529 \text{ nm}$, CIE (0.169, 0.772)) a FOM of $5.1 \times 10^4 \text{ s}^{-1}$ would be required, and for red ($\lambda_{\text{max}} = 650 \text{ nm}$, CIE (0.708, 0.292)) a FOM of at least $1.3 \times 10^5 \text{ s}^{-1}$ is needed. “

We have also added an explanation of how the above target values were calculated in the methods section on page 20-21 of the manuscript.

8) Fig. 1c is an important summary. Fig. 3d is suggestive that $k_{\text{RISC}}/k_{\text{ISC}}$ shows constant values below $\Delta E_{\text{st}} = 0.1 \text{ eV}$.

We agree this is an interesting result. It is useful for the field as it suggests there is no need to reduce ΔE_{ST} below 0.1 eV. However, we have decided to remove this correlation in our revised manuscript to deliver a more concise argument for FOM.

9) Line 397: in equation (4), $krS/kISC \rightarrow kISC/krS$

10) Line 866: Fig. 4b \rightarrow Fig. 3b

These are helpful corrections.

Referee #2 (Remarks to the Author):

The presented manuscript analyses very important and widely accessed topic of roll-off in OLED devices, limiting its practical applicability. A number of TADF compounds with the corresponding OLED devices were studied seeking to find a dominating parameters, describing the potential extent of roll-off in OLED device. Authors developed a model providing relation of roll-off parameter J_{90} with relaxation parameters of singlet and triplet population in TADF materials.

Authors correlate the proposed Figures of merit (FOMS) to the values obtained for 66 devices published by other authors. The results however are widely spread.

- Three correlations are discussed, namely J_{90} with $kRISC$, FOM1 and FOM2, described by correlation coefficients. Since the sample size is statistically not very large and strongly spread, the obtained Spearman correlation coefficients of 0.638, 0.680 and 0.700 looks very similar and it is very difficult to observe any improvement of FOM2 as compared to simple $J_{90} \sim kRISC$ comparison. In order to prove the excellence of FOM2 over other parameters, broader statistical analysis should be performed, indicating that there is a statistical significance in the J_{90} dependences with all three other parameters. Moreover, the spread of data should also be described by statistical parameters.

As explained above the main value of our work is bringing an analysis of the dynamic equilibrium to guide materials development. The proposed figure of merit guides the best efficiency roll-off that could be achieved after optimising a device. We accept that we need to explain this better i.e. that the figure of merit guides materials development and that it was arrived at by considering the physics of the dynamic equilibrium, not by looking at many combinations of rate constants.

We believe the best way to see the improvement is in figure 4, where the solid circles show our FOM, and the grey circles show using $kRISC$ as an alternative. It can be seen the spread in the x direction is reduced by two orders of magnitude for our FOM.

As for the data pool, it is limited by the reported data, which require accurate estimation of J_{90} , k_{RISC} and k_{ISC} , simultaneously. In this context the 66 papers we have analysed provide a significant and important data set. Actually, one purpose of the present study is to call on all researchers to provide more data on the roll-off behaviour.

Figure 4 of the submitted manuscript. Black dots show our figures of merit; grey dots show k_{RISC} . It can be seen that there is a much wider spread of k_{RISC} . Red crosses identify TADF molecules containing heavy atoms that enhance SOC, which leads to an increase in k_{RISC} .

- The impact of material degradation is not properly addressed in the study. That brings very large concern on the validity of J_{90} values used.

We are using data published in reputable journals. Authors should not have published efficiency roll-off data that was not reproducible. By taking J_{90} rather than (for example) J_{50} we are using measurements close to peak efficiency, which is usually a modest current density and less susceptible to degradation than (for example J_{50}). Overall we think it is reasonable to use published data at face value for our analysis.

- The main and totally undiscussed issue is conformational disorder in the solid-state. As it was shown previously, solid surrounding leads to conformational distribution of TADF molecules, enabling the dispersion of singlet-triplet energy gaps. Typical TADF compound may have singlet-triplet energy gap distribution spanning for nearly 100 meV, while the spectroscopically assessed value is just an average of all gaps (see [10.1021/acs.jpcc.9b08269](https://doi.org/10.1021/acs.jpcc.9b08269)). This leads to remarkable variation of rISC values for the same TADF compound, spanning over several orders of magnitude (see [10.1021/acs.jpclett.2c01864](https://doi.org/10.1021/acs.jpclett.2c01864)). As the delayed fluorescence decay in films usually is strongly multiexponential, only very rough fitting can be performed, resulting in the average DF decay time, as well as only rough average of rISC rate. Moreover, weak DF decay usually lasts up to ms time-range, making it very difficult to assess it spectroscopically (see [10.1039/d3tc00482a](https://doi.org/10.1039/d3tc00482a), [10.1103/PhysRevB.68.075208](https://doi.org/10.1103/PhysRevB.68.075208)). Usually, only simple TSCPC or streak

cameras are used, assessing only the initial DF decay. In such case, large errors (e.g. order of magnitude) in estimation of rISC rates occurs (see 10.1021/acs.jpca.0c10391).

The referee is suggesting that published values of k_{RISC} and k_{ISC} may not be reliable. Whilst in some cases this may be true, the objective way to proceed is to include published data at face value. It can be expected that general trends can emerge even if some measurements are incorrect. We are trying to achieve a high-level description based on the physics of the equilibrium. Nevertheless we agree conformational disorder is an important issue and so we added a paragraph to mention it as a possible source of spread of data at the top of page 11:

“We also note that the kinetics of thin films can result in a multi-exponential transient PL, which is caused by conformational disorder⁴². Such a decay can be analysed using a Laplace transformation of the three-level kinetics of each conformer⁴³. Our analysis does not include conformational disorder but could be applied in a similar manner to the analysis of multiexponential transient PL caused by conformational disorder.”

• P.34 Extended Data Fig. 1 a, c, and d - errors in delayed fluorescence lifetime measures and scales.

Thank you for the helpful comment on the typographical mistake in the unit of the delayed fluorescence lifetimes.

Summarizing, though the study is important, however it is based on very diverse data with hardly ensured validity. And above all, no statistical proof of superiority of FOM2 was given. This manuscript cannot be published in Nature.

We hope the introductory explanation of the underlying physical insight, the clarification that our FOM is guiding best possible performance from a material, and the improvement clearly visible in figure 4 (reproduced above) address these concerns.

Referee #3 (Remarks to the Author):

Through summarizing and analyzing previously reported data of TADF-OLED devices and photo-physics, Samuel, Zysman-Colman, and co-workers proposed figures of merit (FOMs) for efficiency roll-off, and discussed correlations between FOMs and J90. Although these FOMs could be useful for TADF-OLED community, obviously it is too specific to meet broad readership of Nature. Here, I also have some comments and questions that need to be addressed by authors:

(1) the number of the used data is small. Thus, the reliability of their conclusions could not be guaranteed. And, all used data in this paper should be listed in SI.

In our data selection session in Methods, we have elaborated the procedures, which we believe can largely guarantee an acceptable reliability and consistency of the data in the present study. Actually, the clear correlations in Fig. 4 and Extended Data Fig. 1 well support our data selection procedure. Here, we want to say that although the sample size is limited (66 papers since 2012), the conclusions are reliable and of significance to the new molecule development. At present all references used are in the methods section (following advice from the *Nature* editor to put them there rather than in SI). We publish all underlying data on a data repository that will be referenced in the paper.

(2) rates of RISC and ISC cannot be measured directly, but rather fitted. The spread of data points in Fig. 4 could be induced by inaccurate fitting of k_{RISC} and k_{ISC} .

The estimation of RISC and ISC has an uncertainty caused by the treatment of non-radiative decay, but is reduced for emitters with higher PLQY (see 10.1021/acs.jpca.1c04056). For the selected devices, PLQY exceeds 60% and thus the uncertainty is reduced. As noted above, we do feel we should use reported values at face value, but have added a comment on page 10 about possible causes of spread of data:

“It should also be noted that practice for determining rate constants varies, which could also contribute to the spread.”

(3) I understand the point made by authors. If we directly use delayed lifetime as FOM including both effects of RISC and S1-state radiative decay, what happens? In fact, delayed lifetime has been used as a good descriptor to screen TADF emitters, see literature *Nature Mater* 15, 1120–1127 (2016).

We agree delayed fluorescence lifetime is a good descriptor of efficiency roll-off, and included it in extended data figure 1a. For these data the magnitude of the Spearman correlation coefficient 0.685 i.e. similar to our FOM. So as delayed fluorescence lifetime is regarded as a good predictor, our FOM should also be regarded as good predictors. Our FOM has an excellent correlation with delayed lifetime as shown in Extended Data Fig. 1.

However, using delayed fluorescence to screen materials would require making and measuring them, and the literature suggests that shortening the lifetime should be done by increasing k_{RISC} . The advantage of our FOM is that it is formulated in terms of rate constants that can be related to aspects of molecular design, and so can guide molecular design for improved efficiency roll-off.

(4) in Figure 1, it is unfair to compare EQE values of OLEDs emitting different colors at the same luminance (1000 nits), as the same luminance may correspond to different photon counts.

As displays run at a given brightness, we believe it is appropriate to compare at the same brightness. Each colour of emission is shown separately in the plot so a reader can look at the behaviour of each colour separately if they wish.

Concluding Remarks

As explained above we believe we could have made it clearer that our FOM is founded on the physics of the dynamic equilibrium that underpins TADF, and that it is a guide for materials development showing the best that can potentially be achieved with a material. We have now distinguished more clearly between “intrinsic” i.e. materials properties affecting efficiency roll-off and “extrinsic” i.e. device properties. We note that photoluminescence quantum yield measurements similarly only considered materials properties and yet proved a very powerful guide to advance OLEDs. We also note that figures of merit proved transformational in the field of nonlinear optics, and that TADF materials is now a very large field requiring such guidance. Inspired by the referees’ comments we have extensively revised the paper to take account of their comments and explain our approach more clearly.

Reviewer Reports on the First Revision:

Referees' comments:

Referee #1 (Remarks to the Author):

The basic concept of the authors is reasonable and I think there is no problem in this regard. The subject of this study, efficiency roll-off, is not too specific, and I think it is an important issue that deserves to be published in Nature.

On the other hand, I think that FoM proposed in a high-level journal like Nature should be an irreplaceable indicator that all researchers in the field will continue to use in the future when developing materials. For this purpose, we still need a stronger correlation between FoM and device lifetime (J90).

I would like to make a few comments as follows.

1) Regarding the design guides:

I have been taken to have suggested $k_r S > k_{ISC} > k_{RISC}$ (I did write that), but the basic idea is the same as that of reviewer 3, shortening the delayed fluorescence lifetime. As I wrote last time "As a result, it has become clear that increasing k_{RISC} alone is not enough to further improve efficiency roll-off, and other processes need to be considered. Researchers in this field now believe that in addition to achieving large $k_r S$ and large k_{RISC} simultaneously, a k_{ISC} smaller than $k_r S$ is important to avoid exciton return to T1," the simultaneous realization of large $k_r S$ and large k_{RISC} is essentially equivalent that of large k_{delayed} (although more precisely k_{ISC} is also involved).

The authors state that "Most researchers understand delayed fluorescence lifetime should be shortened, but believe this should be achieved by increasing the rate of RISC." But, only an increase in k_{RISC} is not sufficient.

2) The FoM proposed by the authors ($= k_r S \times K_{eq}$, from equation (3) in the main text) is a similar indicator to k_{delayed} under certain conditions. In this sense, there should be no problem in assuming $\text{FoM} = k_{\text{delayed}}$ (indeed, FoM and k_{delayed} are well correlated as shown in Extended Data Fig. 1c). From equation (2) in the text, K_{eq} can be calculated from several rate constants, so if all the rate constants related to emission can be predicted, the authors' FoM will be very useful for molecular design. Otherwise, the FoM can only be obtained experimentally. In that case, k_{delayed} is a more direct indicator since the rate constants are calculated from PLQY, k_{prompt} , and k_{delayed} , experimentally.

3) Regarding intrinsic properties (emitter-related) and extrinsic properties (device-related):

As described by the authors, J90 is related not only to the emitter molecule but also to the device structure. This may obscure the correlation between FoM and J90.

However, although the correlation is not clear, it would be very significant if this could be the starting point of an indicator for the development of superior materials and devices. Nothing is perfect from the beginning. But, I would like to request a stronger correlation as described above.

Referee #3 (Remarks to the Author):

The authors have addressed all of my comments. I would recommend it to be accepted as an "Analysis" paper.

Author Rebuttals to First Revision:

A Figure of Merit for Efficiency Roll-off in TADF-based Organic LEDs

Response to Referees' Comments

We are grateful to the referees for their further work on our paper and that both referees feel we have made substantial progress to improve the paper and address their concerns.

Referee #1 (Remarks to the Author):

The basic concept of the authors is reasonable and I think there is no problem in this regard. The subject of this study, efficiency roll-off, is not too specific, and I think it is an important issue that deserves to be published in Nature.

On the other hand, I think that FoM proposed in a high-level journal like Nature should be an irreplaceable indicator that all researchers in the field will continue to use in the future when developing materials. For this purpose, we still need a stronger correlation between FoM and device lifetime (J90).

I would like to make a few comments as follows.

1) Regarding the design guides:

I have been taken to have suggested $k_{\text{RISC}} \sim k_{\text{ISC}} \sim k_{\text{RISC}}$ (I did write that), but the basic idea is the same as that of reviewer 3, shortening the delayed fluorescence lifetime. As I wrote last time "As a result, it has become clear that increasing k_{RISC} alone is not enough to further improve efficiency roll-off, and other processes need to be considered. Researchers in this field now believe that in addition to achieving large k_{RISC} and large k_{RISC} simultaneously, a k_{ISC} smaller than k_{RISC} is important to avoid exciton return to T1," the simultaneous realization of large k_{RISC} and large k_{RISC} is essentially equivalent that of large k_{delayed} (although more precisely k_{ISC} is also involved).

The authors state that "Most researchers understand delayed fluorescence lifetime should be shortened, but believe this should be achieved by increasing the rate of RISC." But, only an increase in k_{RISC} is not sufficient.

We agree with the referee that an increase in k_{RISC} alone is not sufficient and comment on this on page 6 of the manuscript where we say " k_{RISC} alone is inadequate as a predictor of efficiency roll-off."

2) The FoM proposed by the authors ($= k_{\text{RISC}} \times K_{\text{eq}}$, from equation (3) in the main text) is a similar indicator to k_{delayed} under certain conditions. In this sense, there should be no problem in assuming $\text{FoM} = k_{\text{delayed}}$ (indeed, FoM and k_{delayed} are well correlated as shown in

Extended Data Fig. 1c). From equation (2) in the text, K_{eq} can be calculated from several rate constants, so if all the rate constants related to emission can be predicted, the authors' FoM will be very useful for molecular design. Otherwise, the FoM can only be obtained experimentally. In that case, k_{delayed} is a more direct indicator since the rate constants are calculated from PLQY, k_{prompt} , and k_{delayed} , experimentally.

We agree delayed fluorescence lifetime is a good descriptor of efficiency roll-off for materials with very high PLQY, and included it in Extended Data Fig. 1a. However, it does not indicate how to design materials with shorter delayed fluorescence lifetime. Much of the literature suggests that shortening the lifetime should be done by increasing k_{RISC} , and the point of our article is to propose a better approach taking account of the physics of the dynamic equilibrium under electrical excitation. In any case, using delayed fluorescence lifetime to screen materials would require making and measuring them. The advantage of our FOM is that it is formulated in terms of rate constants that can be related to aspects of molecular design, and so can guide molecular design for improved efficiency roll-off. We envisage it potentially being used in two ways. One is that the rate constants are all calculable so it encourages *in silico* screening. The other is that researchers developing materials are likely to have intuition about how to manipulate the rate constants.

3) Regarding intrinsic properties (emitter-related) and extrinsic properties (device-related):

As described by the authors, J90 is related not only to the emitter molecule but also to the device structure. This may obscure the correlation between FoM and J90.

However, although the correlation is not clear, it would be very significant if this could be the starting point of an indicator for the development of superior materials and devices. Nothing is perfect from the beginning. But, I would like to request a stronger correlation as described above.

We appreciate the referee's recognition that nothing is perfect from the beginning, and really hope our proposed figure of merit will advance the field both directly and by stimulating studies and debate of how to overcome efficiency roll-off. The correlation that we obtain is higher than for the referee's preferred measure of delayed fluorescence lifetime, and so is sufficient to be a guide. We have introduced a subsection on "Other Factors Affecting Efficiency Roll-off" where we explain about intrinsic and extrinsic properties, as the referee suggests. We are pleased to see the referee agrees that it would be very significant to have a starting point of an indicator for the development of superior materials and devices.

Referee #3 (Remarks to the Author):

The authors have addressed all of my comments. I would recommend it to be accepted as an "Analysis" paper.